# Generation of Mutants from the 57B Region of *Drosophila melanogaster*

**DOI:** 10.3390/genes14112047

**Published:** 2023-11-06

**Authors:** Eva Louise Steinmetz, Sandra Noh, Christine Klöppel, Martin F. Fuhr, Nicole Bach, Mona Evelyn Raffael, Kirsten Hildebrandt, Fabienne Wittling, Doris Jann, Uwe Walldorf

**Affiliations:** 1Developmental Biology, ZHMB (Center of Human and Molecular Biology), Saarland University, Building 61, D-66421 Homburg, Germany; 2Zoology & Physiology, ZHMB (Center of Human and Molecular Biology), Saarland University, Building B2.1, D-66123 Saarbrücken, Germany; 3Helmholtz Institute for Pharmaceutical Research Saarland (HIPS), Saarland University, Building E8.1, D-66123 Saarbrücken, Germany; 4Medical Biochemistry & Molecular Biology, ZHMB (Center of Human and Molecular Biology), Saarland University, Building 45.2, D-66421 Homburg, Germany

**Keywords:** 57B mutants, 57B deletions, gene targeting, CRISPR/Cas9

## Abstract

The 57B region of *Drosophila melanogaster* includes a cluster of the three homeobox genes *orthopedia* (*otp*), *Drosophila Retinal homeobox* (*DRx*), and *homeobrain* (*hbn*). In an attempt to isolate mutants for these genes, we performed an EMS mutagenesis and isolated lethal mutants from the 57B region, among them mutants for *otp*, *DRx*, and *hbn*. With the help of two newly generated deletions from the 57B region, we mapped additional mutants to specific chromosomal intervals and identified several of these mutants from the 57B region molecularly. In addition, we generated mutants for CG15651 and RIC-3 by gene targeting and mutants for the genes CG9344, CG15649, CG15650, and ND-B14.7 using the CRISPR/Cas9 system. We determined the lethality period during development for most isolated mutants. In total, we analysed alleles from nine different genes from the 57B region of *Drosophila*, which could now be used to further explore the functions of the corresponding genes in the future.

## 1. Introduction

*Drosophila melanogaster* is one of the most powerful model organisms in developmental biology. Due to its genetics and various available techniques, it could also be used to analyse conserved genes associated with human diseases [1]. A major breakthrough came with the sequencing of the *Drosophila* genome and the annotation of its 13,600 genes [2,3]. Attempts to determine the expression patterns of all genes systematically by in situ hybridization and microarray analyses were initiated [3].

Even the analysis of evolutionary processes was possible after having sequenced the whole genomes of in total 12 different *Drosophila species* [4]. The ultimate goal would be of course to analyse the function of all the *Drosophila* genes. Therefore, systematic approaches were made to downregulate *Drosophila* genes with the use of RNA interference [5]. Transgenic strains were generated which enable the downregulation of most genes by RNAi [6,7] using the UAS/Gal4 system [8]. For such experiments large collections of Gal4 strains are available. For example, in one approach putative enhancer regions of 925 genes with a known expression or function in the adult brain were cloned in front of the Gal4 gene [9], 7000 transgenic fly strains were established and the expression patterns in different stages and tissues analysed using reporter genes [10,11,12]. Additional Gal4 strains were made in a complementary approach [13]. In general, this downregulation of genes using RNAi shows effects, but often the genes are not completely inactivated [6].

The UAS/Gal4 system could also be used to analyse gene functions through ectopic expression in tissues where the genes are normally not expressed, here the ectopic expression of Eyeless generating eye structures on the antennae, wings, and legs is a classic example [14]. For these types of experiments, more than 1000 UAS-ORF fly strains are available [15] and additional ones can be made using a high-throughput cloning protocol [16].

Under certain circumstances, mutants of genes would be preferred to the techniques mentioned above. These mutants could be made by forward and reverse genetics approaches. In forward genetic approaches mutants can be made via the generation of EMS (ethyl methanesulfonate)-induced mutations in genes [17], which have to be identified later through detailed mapping experiments and molecular analysis. This method was used in classical screens to identify all genes necessary for pattern formation in the *Drosophila* embryo [18,19,20,21]. The mapping of EMS-induced mutants to specific chromosomal regions can be made with the use of deficiency strains. Therefore a lot of deficiency strains are available as deficiency kits [22] covering most regions of the *Drosophila* genome. Today molecularly defined deficiencies are available which were generated with the help of transposons and enzyme-based recombination [23,24].

Transposons can also be used to inactivate genes, if they are integrated in genes or in important control regions [25]. In a genome wide effort, to inactivate genes and similarly having the possibility to generate defined deficiencies, more than 20,000 transposon insertions were generated [24] and their integration sites were mapped cytologically or molecularly using inverse PCR experiments [26]. As transposons P-elements and piggiBac transposons were used because they have different target site specificities. In about 20% of the cases, these transposon insertions inactivated genes and generated recessive lethal mutants. Since both the EMS mutagenesis as well as transposon integrations are random, these days reverse genetic approaches are more often used to select specific genes and to mutate them directly. This is possible using two different techniques. The first technique is gene targeting, which was first shown in mice [27] and used to make well-defined deletions in coding regions of genes [28] or in enhancer regions [29]. Later on, gene targeting was also established in *Drosophila* [30,31,32]. A more recently developed technique to mutate genes is the CRISPR/Cas9 system [33,34] which also functions in *Drosophila* [35,36,37]. Through the selection of specific target sites double-strand breaks can be made by Cas9 which will be repaired by the organism but often in a way that small deletions or insertions (indels) are made which are not predictable in size, but could of course change open reading frames and thereby inactivate genes.

Summarized our previous work, we isolated three homeobox genes from the 57B region, the gene *orthopedia* (*otp*) [38,39], *Drosophila Retinal homeobox* (*DRx*) [40,41], and *homeobrain* (*hbn*) [42,43]. All genes are located in a 77 kb interval and build a cluster of homeobox genes showing very interesting expression patterns during development mainly in the nervous system [43,44,45,46]. To analyse the function of these genes we isolated mutants from the 57B region performing a classical EMS mutagenesis experiment using a deficiency uncovering the 57B region and succeeded in the isolation of lethal mutant alleles for all three genes [39,43,47]. 

In addition to these mutants, we isolated other lethal mutants from the 57B region which were not analysed yet. In this paper, we show the analysis of some of these mutants from the EMS-induced mutagenesis. In addition, we generated new deletions of the 57B region and new mutants made by more advanced techniques like gene targeting and using the CRISPR/Cas9 system. Totally we analysed alleles from nine different genes from the 57B region of *Drosophila*, which could now be used to further explore the functions of the corresponding genes and the 57B region in more detail.

## 2. Materials and Methods

### 2.1. Fly Strains

Fly strains ordered from Bloomington Drosophila Stock Center (BDSC) are given with BL-numbers and strains ordered from the Vienna Drosophila Resource Center (VDRC) are given with their VDRC-numbers. Other strains are given with the direct source.


*The following fly strains were used:*
y[1] w[*]; P{w[+mC]=Act5C-GAL4}17bFO1/TM6B, Tb[1] (BL 3954);y[1] w[1118]; P{ry[+t7.2]=70FLP}23 P{v[+t1.8]=70I-SceI}4A/TM3, Sb[1] Ser[1] (BL 6935);w|1118]; Df(2R)Exel6072, P{w[+mC]=XP-U}Exel6073/CyO (BL 7554);w[1118]; Df(2R)Exel7166, P+PBac{XP5.WH5}Exel7166/CyO (BL 7998);w[1118]; wg[SP-1]/CyO; sens[Ly-1]/TM6B, Tb[1] (BL 8136);P{y[+t7.7]=nos-phiC31\int.NLS}X, y[1] sc[1] v[1] sev[21]; P{y[+t7.7]=CaryP}attP2 (BL 25710);y [1] M{w[+mC]=nos-Cas9.P}ZH-2A w[*] (BL 54591);P{PZ}l(2)07206 (21,057,826) [48];P{PZ}l(2)07806 (20,975,585) (BL 12342);P{GSV6}GS10771 (20,995,922) (No 202-812 *Drosophila* Genetic Resource Center, Kyoto);PBac{WH}f03917 (20,882,698),P{PZ}d11699 (20,975,379),PBac{WH}f04151 (21,030,314) (Exelixis *Drosophila* stock collection, Harvard Medical School).



*RNAi strains:*
CG15649:w[1118]; P{GD15200}v42358/TM3 (VDRC 42358)P{KK113097}VIE-260B (VDRC 103443)CG15650:y[1]v[1]; P{TRIP.HMJ22721}attP40 (BL 60430)w[1118]; P{GD5930}v13872 (VDRC 13872)P{KK107072}VIE-260B (VDRC 101188)CG15651:y[1]v[1]; P{TRIP.JF03295}attP2 (BL 29616)y[1]v[1]; P{TRIP.HMC03463}attP40 (BL 51889)w[1118]; P{GD2117}v37206P{KK101592}VIE-260B (VDRC 105421)ND-B14.7:w[1118]; P{GD4517}v31562P{KK109518}VIE-260B (VDRC 110295)CG9344:y[1]v[1]; P{TRIP.HM05211}attP2 (BL 29532)y[1]v[1]; P{TRIP.HMJ21492}attP40 (BL 54798)w[1118]; P{GD13733}v23689 (VDRC 23689)


### 2.2. Generation of 57B Deletion Strains

To generate molecular-defined deletions within the 57B region, we used existing transposon insertion strains following exactly the method described in [23]. In short, the respective transposons bearing FRT sites have to be in trans, and through appropriate crossings, a heat-shock-induced Flippase mediates a recombination event via the FRT sites to generate the deletion. For the deletion of Df(2R)57B3-5 the transposon strains PBac{WH}f03917 and P{PZ}d11699 were used, for Df(2R)57B5-13 the transposon strains P{PZ}d11699 and PBac{WH}f04151. After the successful recombination event, a hybrid transposon is created consisting of parts of both original transposons instead of the deleted region. To prove if the recombination event happened as predicted two-sided PCR reactions were performed with primers binding in the transposons and the neighboring genomic regions. For Df(2R)57B3-5 the genomic primer f03917-5 (5′-CACGCCGACCTCATGTCCTG-3′) and the piggiBac primer 5R1 (5′-TGACACTTACCGCATTGACA-3′) were used for one side of the deletion, for the other side the P-element primer XP-3SEQ (5′-TACTATTCCTTTCACTCGCACTTATTG-3′) and the genomic primer d11699-3 (5′-GTGGGCTCCCCCCACAAGGTG-3′) were used. For deletion Df(2R)57B5-13 the genomic primer d11699-5 (5′-CAGGACTTTACTTTCCCCTGAG-3′) and the P-element primer Plac1 (5′-CACCCAAGGCTCTGCTCCCACAAT) were used for one side and the piggiBac primer 3F1 (5′-CAACATGACTGTTTTTAAAGTACAAA-3′) and the genomic primer f04151-3 (5′-ATATAGCAAGGTCCACAGTTCGG-3′) for the other side. The final strains were balanced over CyO and used for the complementation assays.

### 2.3. Sequence Analysis of Mutant Strains with Point Mutations

To identify sequence alterations in the mutant strains, amplifications of the coding region were performed using genomic DNA from the heterozygous mutant fly stocks. For polymerase chain reactions Taq Polymerase from Thermo Fisher Scientific (Waltham, MA, USA) was used according to the supplier’s instructions. PCR products were sequenced by Starseq (Mainz, Germany). The regions showing sequence alterations compared with the wild-type sequence were again PCR-amplified using more closely located primers. In addition, the PCR products were subcloned into the TOPO vector pCR2.1 (Thermo Fisher Scientific, Waltham, MA, USA), and at least 10 individual clones from PCR product cloning were checked by sequencing. Since the mutant DNA was generated from heterozygous flies approximately 50% of the clones showed the wild-type sequence and 50% the altered sequence due to the point mutation.

### 2.4. Generation of CG15651 and RIC-3 Mutants by Ends-Out Homologous Recombination

Donor gene targeting constructs were generated as previously described [32]. For the CG15651 donor construct the primers 15651GT1 (5′-CGTACGATTTCTCCTGGATACCATTCCTTC-3′) and 15651GT2 (5′-GGCGCGCCAAGAACATCCTGCAGAAGATTCTG-3′) were used in a PCR with Pacman 83M21 DNA [49] as template to amplify a 3.6 kb fragment (15651-I) with *Bsi*WI and *Asc*I sites (underlined). The fragment was cut with *Bsi*WI and *Asc*I and cloned into the pW25 targeting vector (*Drosophila* Genomics Resource Center, (DGRC), Stock 1166) to generate pW25/15651-I. Next, primers 15651GT3 (5′-GGTACCTTCCTCCCAGGATATGTGACCG-3′) and 15651GT4 (5′-GCGGCCGCACAGATCGGAGCTCTTCTCGTG-3′) were used in another PCR with 83M21 DNA to add *Kpn*I and *Not*I sites (underlined) to a 3.4 kb fragment (15651-II) which was then cut with *Kpn*I and *Not*I and cloned into the corresponding sites of the vector pW25/15651-I to generate the final construct CG15651-pW25. For the RIC-3 donor construct the primers RIC-3GT3 (5′-GGTACCGCCGATTCTAGAACGAAAGACTTCG-3′) and RIC-3GT4 (5′-GCGGCCGCAATTTGGACTGAAATCGAAGAGGG-3′) were used in a PCR with Pacman 131C22 DNA [49] as a template to amplify a 3.8 kb fragment (RIC-3-II) with *Kpn*I and *Not*I sites (underlined). The fragment was cut with *Kpn*I and *Not*I and cloned into the pW25 targeting vector (DGRC) to generate pW25/RIC-3-II. Next, primers RIC-3GT1 (5′-GGCGCGCCGTTGGATATGACATGACCGCAG-3′) and RIC-3GT2 (5′-GGCGCGCCTATCTGGATTCGTTTGGCTATGG-3′) were used in another PCR with 131C22 DNA to add *Asc*I sites on both ends (underlined) to a 3.4 kb fragment (RIC-3-I) which was then cut with *Asc*I and cloned into the corresponding site of the vector pW25/RIC-3-II to generate the final construct RIC-3-pW25.

Transgenic fly lines with the donor targeting constructs were made by germline transformation using standard techniques [50]. Donor targeting strains were crossed to *yw*; *70FLP*, *70I-SceI*, *Sco/CyO* flies and the F1 progenies were heat-shocked at 37 °C for 1 h on days 3, 4, and 5 after egg laying. The resulting targeting flies were selected according to their eye colour, balanced, and the correct gene targeting event was molecularly verified by PCR amplification with genomic DNA of these flies as a template and primers CG34114seq (otp; 5′-CGTCCGGCACTTTGGCACG-3′) and P2 (pW25 vector; 5′-CGTGCTCATCGCGAGTACG-3′) or P3 (pW25 vector; 5′-GAGTGCCGTTTACTGTGCG-3′) and CG34115seq (otp; 5′-GAGCAGCCCAGATTCCATGC-3′). After the verification step, the *white* gene which was replacing the deleted genomic regions in both strains was removed by Cre recombinase using the two adjacent loxP sites to generate the final targeting strains. Instead of the deleted regions, a single loxP site is then left in the respective loci.

### 2.5. gRNA Design, Cloning of CRISPR Constructs, and Analysis of CRISPR/Cas9-Induced Mutations

Target sites were selected to mediate Cas9-induced cleavage close to the 5′ end of the coding sequence. To avoid off-target cleavage, target sites were selected using the CRISPR target finder [51]. Target site oligos were annealed and cloned in the vector pCFD3-dU6:3 [52] cut with *Bbs*I according to [53]. The oligos used are: CRISPR-CG9344A (5′-GTCGTCGCCGTAAAGCTGAACAA-3′) and CRISPR-CG9344B (5′-AAACTTGTTCAGCTTTACGGCGA-3′) for CG9344; CRISPR-CG15649A (5′-GTCGCGGCCATAAAGACAAATCTC-3′) and CRISPR-CG15649B (5′-AAACGAGATTTGTCTTTATGGCCG-3′) for CG15649; CRISPR-CG15650A (5′-GTCGCCACGCTGCACGATCTGCC-3′) and CRISPR-CG15650B (5′-AAACGGCAGATCGTGCAGCGTGGC-3′) for CG15650; CRISPR-ND-B14.7A (5′-GTCGAAATACTACGACCATCCCGA-3′) and CRISPR-ND-B14.7B (5′-AAACTCGGGATGGTCGTAGTATTT-3′) for ND-B14.7. Correct constructs were identified by PCR and verified by sequencing. Transgenic fly lines were generated by PhiC31-integrase mediated transformation [54] in the attP2 site (68A4) using strain BL 25710 and standard techniques [50]. Transformants with the individual CRISPR constructs were crossed to a nanos-Cas9 strain (BL 54591) and individual males of the offspring balanced over CyO. From individual strains genomic DNA was prepared and the Cas9 target region was PCR amplified using primers 5′ and 3′ of the target region (primer sequences are available upon request). The PCR products were sequenced by Starseq (Mainz, Germany).

## 3. Results

### 3.1. Isolation of 57B Mutants and Rough Mapping Using Newly Generated 57B Deletions

The 57B region on the second chromosome received the main focus since a cluster of three homeobox genes is located there. The three genes *orthopedia* (*otp*) [38,39], *Drosophila Retinal homeobox* (*DRx*) [40,41], and *homeobrain* (*hbn*) [42,43] are all expressed in the nervous system throughout *Drosophila* development. To isolate mutants for these genes, we planned to generate EMS-induced mutants from the 57B region using the Df(2R)E2 (57B1-57B13,14) [55]. Before starting the mutagenesis experiment we used lethal P-element strains in the genes *exuperantia* (*exu*) (57A9–7A10) [56], *inscuteable* (*insc*) (57B3) [57], and *shotgun* (*shg*) (57B15–B16) [55] to better map the extent of this only cytologically characterized deletion. We determined, that the proximal breakpoint of this deficiency is located between the genes *exu* (57A9–57A10) and *insc* (57B3) and the distal breakpoint distal of the gene *shg* (57B15–B16). We then mutagenized *b pr cn bw* males with EMS using standard protocols [58] and by the appropriate crosses isolated 36 mutants of the 57B region among the offspring of 6278 crosses (see Appendix A). Our mutants represented 20 complementation groups, among them 3 *inscuteable* (*insc*) alleles [57] and 6 *shotgun* (*shg*) alleles [55,59], the two *otp* alleles *otp*^1024^ and *otp*^13064^ [39], the *DRx* allele *DRx*^10155^ [47] and the two *hbn* alleles *hbn*^4028^ and *hbn*^15227^ [43]. These alleles could be identified due to their phenotype, complementation tests with existing mutants, and finally sequencing of the coding region.

Since all the other mutants isolated could not be assigned to known genes, we tried to identify at least some of them. For a first rough mapping, we used already existing molecularly defined deletions of the 57B region and two new deletions which we generated on the basis of already existing P-element and piggiBac strains available [24]. There were two deficiencies already available at that time, Df(2R)Exel7166 deleting 57B3-B5 (20.870.857–21.000.163), a region of 130 kb and Df(2R)Exel6072 deleting 57B16-57D4 (21.056.789–21.250.845), a region of 194 kb. Both of these deletions still leave a gap of 56 kb in between which is not covered. Therefore we generated the deletion Df(2R)57B3-5 using the piggiBac strain PBac{WH}-f03917 (20.882.698) and the P-element strain P{PZ}d11699 (20.975.379) and the deletion Df(2R)57B5-13 using again the P-element strain P{PZ}d11699 (20.975.379) and the piggiBac strain PBac{WH}f04151 (21.030.314) [23]. With the help of these deficiency strains, it was possible to better assign our mutations to defined genomic intervals (Figure 1A). Mutant 8060 could be assigned to Df(2R)57B3-5, mutants 7272 and 3311 to the region deleted in Df(2R)57B5-13 and Df(2R)Exel7166, and mutants 17104, 17141 and 20243 to the region deleted in Df(2R)57B5-13, but not in Df(2R)Exel7166 (Figure 1A).

### 3.2. Identification of EMS-Induced Mutants from the 57B Region

We next looked for P-element strains from the 57B region whose integration points were already known and used them in complementation experiments to identify corresponding mutants from our EMS-mutant collection. The lethal P-element strain P{PZ}l(2)07806 generated in the course of the Berkeley *Drosophila* genome project [48] did not show complementation with our strain 7272 (Figure 1A). At the time when this P-element strain was made, it was not known which gene was inactivated by the insertion, but later CG34115 was identified as the gene that was affected. To identify the exact position of the EMS-induced point mutation in strain 7272 we PCR amplified the coding region including the splice sites of CG34115 and sequenced the PCR products. In strain 7272, the start codon of CG34115 is altered from ATG to ACG, therefore due to the T to C transition the methionine triplet is not present and the open reading frame does not start at the right position (Figure 2A). We called the mutant strain 7272 from now on CG34115^7272^.

The strain P{GSV6}GS10771 (*Drosophila* Genetic Resource Center, Kyoto) which has an insertion in the Prosα3 gene from 57B (originally known as Pros29; [60]) did not show complementation with our EMS-mutant strain 3311 (Figure 1A). To identify the point mutation in strain 3311 we again PCR amplified the coding region including the splice sites of Prosα3 and sequenced the PCR products. We found a sequence alteration in a splice donor site which normally is GT and in strain 3311 is GA (Figure 2B). Since splice sites are highly conserved [61], we believe that this sequence alteration is responsible for the inactivation of Prosα3 and from now on called this new Prosα3 allele Prosα3^3311^.

Three other strains did not show complementation with the lethal P-element strain P{PZ}l(2)07206 mapping in 57B16. The corresponding gene in this region is the *capping protein α* (*cpa*) encoding an actin-binding protein. We analysed the three EMS-mutant strains 3072, 13230, and 18207 molecularly and found a C to T transition in codon 173 resulting in a change from serine to phenylalanine in strain 3072 (Figure 2C). In strain 18207 a G to A alteration results in a stop codon leading to a truncated protein of 174 amino acids instead of 286 amino acids of the wild-type protein (Figure 2C). In strain 13230 we could not identify a sequence alteration.

### 3.3. Generation of CG15651 and RIC-3 Mutants by Gene Targeting

After analysis of all EMS-induced mutants which could be identified by complementation analysis with existing deletion strains, we looked for 57B genes with interesting expression patterns by in situ hybridization and identified CG15651 with a strong expression in the complete brain and ventral nerve cord in the embryo (Berkeley Drosophila Genome Project). RIC-3 on the other hand is of interest since it belongs to a highly conserved gene family [62,63] and is predicted to have an acetylcholine receptor binding activity and seems to be involved in chaperone-mediated protein folding [64]. For these reasons, we decided to make mutants for CG15651 and RIC-3 by gene targeting (Figure 1B). By ends-out gene targeting [30,31,32], we generated the mutant allele *CG15651*^GT^ using the targeting vector pW25 [32] (Figure 3A).

In this mutant a 839 bp deletion starting 32 bp upstream of the 5′UTR and including the first 509 bp of the open reading frame is present. In the case of *RIC-3* the mutant allele *RIC-3*^GT^ has a deletion of 820 bp starting 26 bp upstream of the ATG in exon 2 and including exon 3 and half of exon 4 (Figure 3B). In both strains, the *white* gene which was originally integrated in the locus instead of the deleted gene sequences was later deleted using Cre recombinase via the loxP sites flanking the *white* gene. In the final targeting strains, a single loxP plus some adjacent vector sequences are left in the locus. The correct gene targeting events were verified by PCR amplification over the deleted area and sequencing of the PCR products. Both targeting strains *CG15651*^GT^ and *RIC-3*^GT^ were homozygous viable.

### 3.4. Generation of Mutants Using CRISPR/Cas9

To obtain an idea of which other genes in the 57B region might be candidates for essential genes, we downregulated some of the not yet analysed genes by RNA interference. We used available RNAi strains from the Bloomington *Drosophila* Stock Center (BDSC) and the Vienna *Drosophila* Resource Center (VDRC) and crossed them to an Act5C-Gal4 strain to downregulate the gene of interest in all cells during development. When RNAi strains of the genes CG9344, CG15649, CG15650, and ND-B14.7 were crossed to the Act5C-Gal4 strain and the offspring was analysed, early or late pupal lethality was observed arguing for an essential function of the corresponding genes.

To generate mutants of the genes CG9344, CG15649, CG15650, and also ND-B14.7 which were supposed to be lethal when mutated, we used the CRISPR/Cas9 system [33,34,35]. The location of these genes in the 57B region is shown in Figure 1B. To avoid off-target cleavage of Cas9, we used a CRISPR target finder [33] to identify optimal target sites in the respective genes close to the 5’ end of the coding region to generate N-terminally truncated protein forms of the genes (Figure 4).

The corresponding constructs were made in the pCFD3 vector [52] and transgenic fly strains were generated through PhiC31-mediated integration in the attP2 site on the third chromosome. In these flies the gRNAs were under the control of the *U6:3* promoter. The flies were crossed to a fly strain expressing Cas9 under the control of the *nanos* promotor to obtain efficient germline transmission of the mutations. In most cases, we isolated through the appropriate crosses and balancing of the resulting males between 50 to 100 strains for each gene. Of these strains roughly one-third were homozygous-lethal, and the others were homozygous-viable. To analyse the putative sequence alterations, we started with homozygous lethal strains and amplified the region of interest by PCR and sequenced the PCR products. Due to the heterozygosity of the flies, we obtained, in the region of interest, the wild-type sequence and the mutant sequence on top, but the sequencing peaks could be easily read out by hand or using special computer software.

In the case of CG15649 and CG15650, we often identified the wild-type sequence in homozygous lethal strains pointing out to us, that the lethality was not due to alterations in the gene where it was supposed to be, but rather in an off-target region. We therefore now analysed homozygous viable strains of CG15649 and CG15650. In CRISPR mutants of CG15649 and CG15650, we identified in the region of interest deletions and insertions leading to reading frame changes in most cases, some are indicated in Figure 5. These results clearly indicate that alterations of the open reading frame in CG15649 and CG15650 do not lead to lethality, so both genes are non-lethal genes.

In the case of CG9344 and ND-B14.7, we always identified alterations in the homozygous lethal strains, some of them having alterations in the open reading frame are shown in Figure 5. This pointed out, that in contrast to CG15649 and CG15650, CG9344 and ND-B14.7 represent lethal genes. Finally, we determined the lethality period of CG9344 and ND-B14.7 strains which arise at the larval/pupal stage.

## 4. Discussion

In this paper, we isolated EMS-induced mutants from the 57B region of *Drosophila* and assigned them to specific genes. In addition, we used reverse genetics techniques like gene targeting and the CRISPR/Cas9 system to generate mutants of specific genes of interest within the 57B region and analysed them molecularly.

The analysis of EMS-induced mutants is very time-consuming, since it is first necessary to map the location of the mutation with the help of deficiencies or already existing mutants using complementation experiments. Depending on the length of the genomic interval sequencing experiments can follow. In our case, we already did the screen using a 57B deficiency but also had to map them exactly within the analysed region. These days the identification of such EMS-induced point mutations might also be possible using next-generation sequencing techniques to sequence genomic DNA or RNA [65,66]. We did this for one of our *otp* mutant alleles, but could not unambiguously identify the point mutation since we had not enough reads to see alterations several times.

In the classical *Drosophila* screens by Nüsslein-Volhard and coworkers 26,978 fly strains were generated, among them 600 embryonic lethal mutants representing 120 genes [67], so on average 5 mutant alleles per gene. In our screen we generated 36 mutants representing 20 genes, so on average 1,8 mutants per gene. Nevertheless, we think that we obtained mutants for the lethal genes in the region we analysed. This is supported by the fact that a saturation mutagenesis in the *zeste-white* region which has 42 transcription units resulted in 16 mutants [68], whereas we identified 20 lethal mutants in a region with 37 transcription units. The frequency of alleles identified for a certain gene depends on the gene size serving as a target for a putative sequence alteration, this was also the case in our experiment. Examples are the 6 identified *shg* alleles (protein length 1507 aa), the 3 *insc* alleles (protein length 859 aa), and the 3 *cpa* alleles (protein length 286 aa), whereas most genes where only one allele was identified in our screen code for smaller proteins.

By complementation analysis with existing transposon-induced lethal mutant strains and molecular analyses we identified mutants in CG34115 and Prosα3. CG34115 is expressed in embryonic and larval muscle cells (Berkeley *Drosophila* Genome Project, BDGP) and codes for a protein of 77 amino acids with unknown function. Prosα3 shows in the embryo maternal expression, expression in pole cells, and later on expression in germ cells, head, and trunk mesoderm (BDGP) and codes for a 264 amino acids protein. It is orthologous to human PSMA4 (proteasome 20S subunit α 4) [69] and it is predicted to be involved in proteasomal ubiquitin-dependent protein catabolic processes. A muscle-specific downregulation of proteosomal components, among them Prosα3 shows that proteosomal stress mounts a long-range protective response that delays retinal and brain aging [70].

The *cpa* gene for which we identified three new alleles is coding for the α-subunit of the F-actin capping protein from *Drosophila*. The α-subunit is forming a heterodimer with the β-subunit and mutants of both subunits are known to cause actin accumulation and subsequent retinal degeneration [71]. In the mutant allele *cpa^18207^* a stop codon is introduced leading to a truncated protein of 174 amino acids missing the C-terminus which is important for binding to actin filaments [72]. In *cpa^3072^* amino acid 173 is changed from serine to phenylalanine. These residues are located in a domain that is important for the interaction of the α- and β-subunits of the F-actin capping protein. 3D modelling revealed that the larger space of the phenyl ring compared to the OH-group of serine leads to a change of distances of several adjacent amino acids. Alternatively, serine might be a putative phosphorylation site regulating the activity of the capping protein.

Compared to the analysis of point mutations, the analysis is of course much different if gene targeting or the CRISPR/Cas9 system is used to generate mutants. With gene targeting it is possible to make exactly defined deletions of genomic sequences of a defined gene. Usually, these deleted sequences are replaced by an integrated white gene marker, which could also be excised afterwards using flanking loxP sites leaving back a loxP site and some flanking sequences. In recent years more sophisticated vectors were made which make gene targeting easier and have additional advantages. One such vector is pTV^cherry^ [73], which combines easier cloning of the homology arms for targeting and a better selection of the targeting flies from the donor construct via excision and homologous recombination in the target locus. Another advantage is the integration of an attP site in the locus instead of the deleted sequence, which could be used afterward to reintegrate various DNA sequences like a Gal4 sequence or a reporter gene with the help of specific reintegration vectors [73]. In our hands, this worked pretty well and we generated new mutant alleles for the genes *otp*, *DRx*, *hbn*, and *earmuff* (*erm*) [74] and reintegrated Gal4 into the locus to make Gal4 strains of the corresponding genes [44,45,46,75]. Our targeting efficiencies were in the same range as those reported by [73] between 1/1000 and 1/2000. We encountered only one drawback since one has to take care of the insertion of the donor construct used for gene targeting. Since after the Flippase-induced recombination of the donor construct some P-element sequences are leftover at that position, the integration of the donor in a gene might affect the function of that gene even in the final targeting strain. Therefore, it is advisable to use donor strains with integration sites on other chromosomes with respect to the targeted gene or those where the integration is not critical.

In our targeting experiments, we generated mutants for the genes CG15651 and RIC-3, which were both viable. After we finished our experiments it was published, that CG15651 codes for a Fukutin-related protein [76] and is involved in olfactory memory formation [77]. Human ortholog(s) of CG15651 are implicated in dilated cardiomyopathy and muscular dystrophy [78].

From the RIC-3 gene 19 alternatively spliced isoforms are known in *Drosophila* coding for proteins ranging between 336 and 505 amino acids [63,78]. In mice, it was shown that RIC-3 expression and splicing can regulate the functional expression of nicotinic acetylcholine receptors [79]. Recently, the effects of cofactors RIC-3, TMX3, and UNC-50 on *D. melanogaster* Dα1/Dβ1 nicotinic acetylcholine receptors were analysed [80].

Another technique we used to generate mutants was the CRISPR/Cas9 system. Our experiments were designed according to experiments of [52] using the vector pCFD3-dU6:3. The cloning procedure in the vector is fast and the generation of transgenic flies is very efficient due to the small size of the vector. Also, the generation of Cas9-induced mutations is effective and various mutations could be identified in the coding regions of the four genes we analysed. The only problem in our experiments were strains with mutations in other genes (off-targets). In some cases, we obtained fly strains showing lethality and others not showing lethality. Here we assumed that the lethal strains would be the right ones and the viable strains would have no sequence alteration or only such which do not affect the function of the analysed gene, but this turned out wrong sometimes. We identified sequence alterations in the coding region of the analysed gene in viable flies and no sequence alteration in the lethal strains clearly indicating that in these strains off-target genes are affected. Sequence alterations were as expected deletions and insertions (indels) of various lengths. Also, the target-specific precision of CRISPR-induced editing can vary considerably with some targets showing dominant indels, whereas others show many diverse ones [81], this was also the case for the indels we analysed.

For the genes CG15649 and CG15650 viable mutants were isolated. CG15649 codes for a 119 amino acids protein, CG15650 for a protein of 145 amino acids, both with unknown function. A downregulation of CG15650 shows effects in trichogen cells and in the middle segment of the thorax [82,83].

For the genes CG9344 and ND-B14.7 we isolated lethal mutants. CG9344 is ubiquitously expressed in the embryo, codes for a small 79 amino acids protein with a poly(A)RNA binding domain [84], and has homology to the human LSM6 protein which is involved in splicing [85]. It interacts also with other splicing proteins [86].

ND-B14.7 is coding for the 170 amino acids long NADH dehydrogenase subunit B14.7 [87] which is presumably involved in pain perception [82]. ND-B14.7 and its human ortholog NDUFA11 are known as essential subunits of the mitochondrial respiratory complex I (CI), the largest discrete enzyme of the oxidative phosphorylation system (OXPHOS) in the mitochondrial inner membrane [88,89]. The central roles of CI in metabolism as well as in oxidative stress make dysfunctions of ND-B14.7/NDUFA11, e.g., induced by mutations, a cause for mitochondrial diseases [89]. Previous studies found that the knockdown of NDUFA11 by RNAi in human cell culture led to partially assembled subcomplexes, leading to the assumption of NDUFA11 as an assembly factor in CI [90]. Furthermore, the loss of NDUFA11 in a knockout strain of human cell culture resulted in no expression of CI [91]. As *Drosophila* is a candidate model organism for CI, due to its high evolutionary conservation with the mammalian enzyme, our CRISPR-induced mutations may help to resolve open questions about this complex.

In the meanwhile, the CRISPR/Cas9 system was used for large-scale mutagenesis experiments with the generation of a library of 2600 plasmids and the generation of 1400 fly strains focusing on kinases, phosphatases, and transcription factors [92]. In parallel collections of additional transgenic sgRNA lines were generated by other research groups [93,94]. In addition to Cas9 also other nucleases like Cas12a can be used. Cas12 is as effective as Cas9 but has the advantage of enabling modulation of gene editing by temperature, high activity at 29 °C, and low activity at 18 °C [95].

In summary, we generated 26 mutants for genes in the 57B regions using forward as well as reverse genetic methods. These days the reverse genetic methods might be the choice for the generation of mutants, here mainly the CRISPR/Cas9 system. The advantage of this system is the easier generation of mutants, the availability of libraries, the usage of different CAS enzymes, and the fact that the problem of off-target sites might be not as high as estimated as it was shown in mice [96].

## Figures and Tables

**Figure 1 genes-14-02047-f001:**
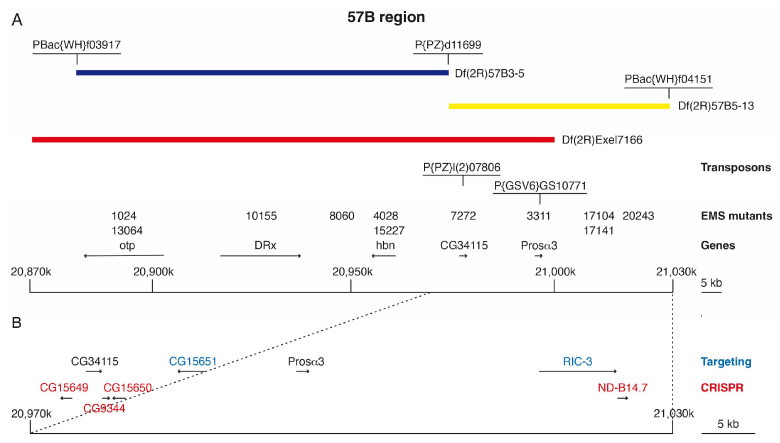
The 57B region. (**A**) Genomic organisation of a 160 kb interval of the 57B region (20,870 k to 21,030 k). The deletions used to map the 57B mutations are indicated as blue, yellow, and red bars. The relevant genes are indicated by their orientation (arrows) and the numbers of the corresponding mutations shown with their position. The mutant 8060 was mapped to the region 57B3-5, and the mutants 17104, 17141, and 20243 were mapped in the region of Df(2R)57B5-13 not overlapping with Df(2R)Exel7166 (the indicated positions were elected by chance). The integration sites of the transposons used in complementation experiments and to generate the deletions Df(2R)57B3-5 and Df(2R)B5-13 are shown above the mutants. (**B**) Enlargement of a 60 kb Interval of the 57B region (20,970 k to 21,030 K) showing the location of the genes analysed by gene targeting in blue and using the CRISPR system in red.

**Figure 2 genes-14-02047-f002:**
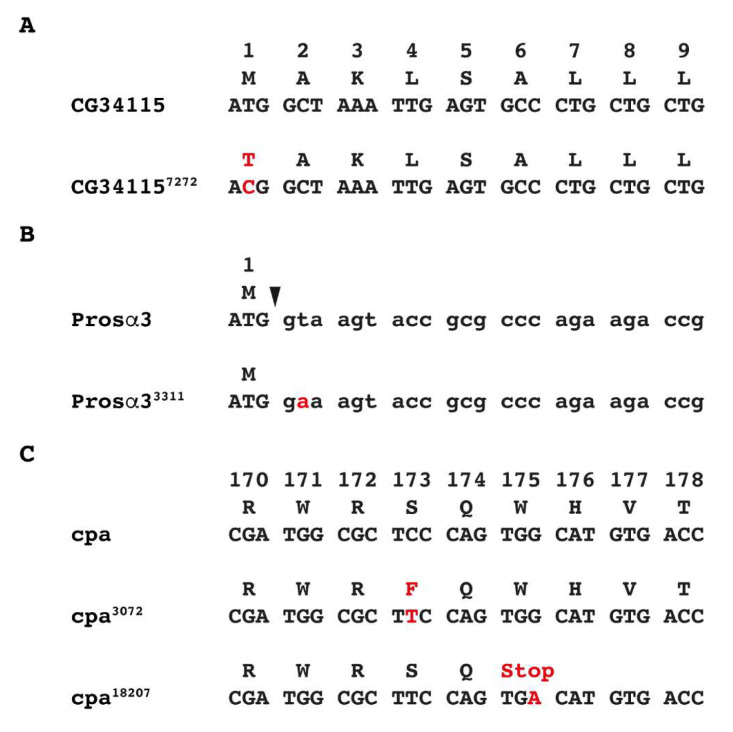
EMS-induced mutations in genes of the 57B region. (**A**) CG34115 nucleotide and amino acid sequences in wild-type and mutant DNA. EMS induces a T to C transition in the start codon of the *CG34115*^7272^ coding region. (**B**) Prosα3 nucleotide and amino acid sequences in wild-type and mutant DNA, here a T to A mutation is changing the splice site in Prosα3^3311^; the splice site is indicated by an arrowhead. (**C**) Cpa nucleotide and amino acid sequences in wild-type and two mutant DNAs. In *cpa*^3072^ a C to T transition is changing amino acid 173 from serine to phenylalanine. In *cpa*^18207^ a A to G transition results in a stop codon at position 175.

**Figure 3 genes-14-02047-f003:**
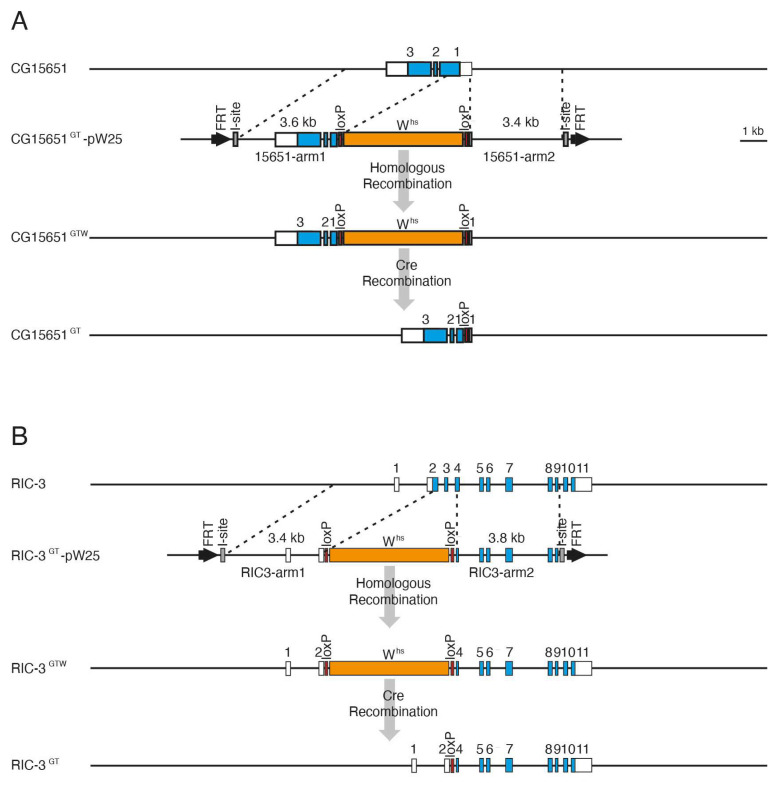
Gene targeting of CG15651 and RIC-3. (**A**) At the top, the genomic organization of the CG15651 gene is shown with exons indicated by boxes, the untranslated regions are shown in white, translated regions in blue. The CG15651 gene targeting construct in the targeting vector pW25 (CG15651^GT^-pW25) is shown below (FRT, FLP recombination target sequences; I-site, I-SceI recognition site; loxP, loxP site; whs, white gene with hsp70 promoter). Further down the CG15651 genomic region after the gene targeting event is indicated (CG15651^GTW^); here part of exon 1 (containing the translation start) is replaced by the white gene of the targeting construct. The white gene is then removed by Cre recombinase and a single loxP is left in the locus instead of the deleted region. (**B**) As shown for CG15651 in (**A**) the respective steps are indicated for the RIC-3 gene targeting. Here parts of exon 2 (containing the translation start) and exon 4 are deleted as well as the complete exon 3.

**Figure 4 genes-14-02047-f004:**
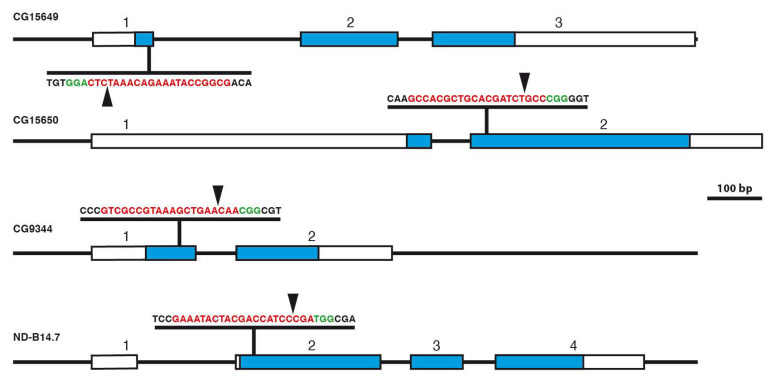
CRISPR target sites in CG15649, CG15650, CG9344, and ND-B14.7. The genomic organisation of genes CG15649, CG15650, CG9344, and ND-B14.7 is shown. Exons are indicated by boxes and numbers, the untranslated regions are shown in white, and the translated regions are in blue. The sequence and position of the gRNAs are indicated (gRNA sequence in red, PAM motif in green), and the Cas9 cleavage sites were indicated by arrowheads.

**Figure 5 genes-14-02047-f005:**
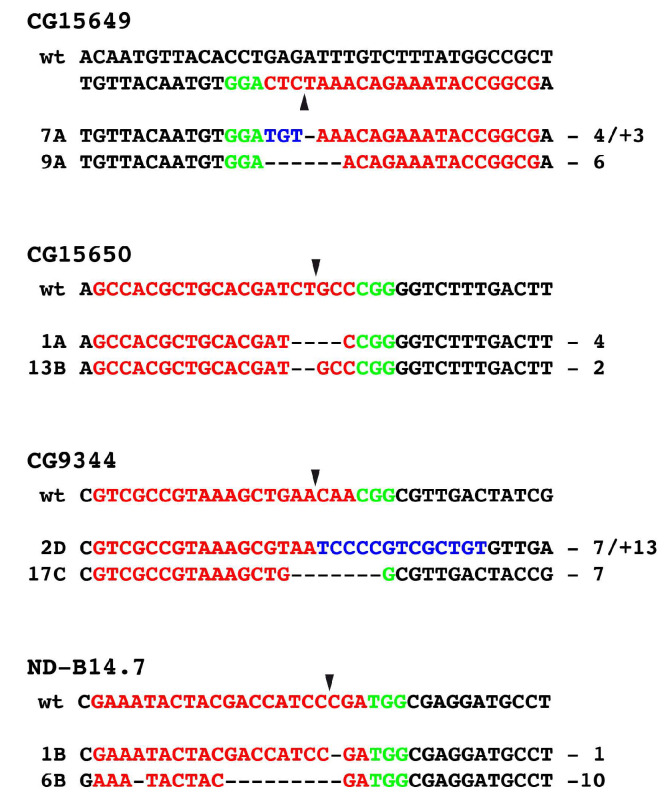
CRISPR/Cas9-induced germline mutations in CG15649, CG15650, CG9344 and ND-B14.7. Examples of CRISPR/Cas9-induced germline mutations in the respective genes. Protospacer sequences are shown in red, and the PAM sequence in green. The position of the Cas9 cleavage site is shown by an arrowhead. Deletions are indicated by hyphens, additional bases are shown in blue.

## Data Availability

Not applicable.

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
