# Peer review of "Generation of Mutants from the 57B Region of Drosophila melanogaster"

_genes, 2023, doi:10.3390/genes14112047_

Round 1

Reviewer 1 Report

Comments and Suggestions for Authors

This paper by Walldorf and collaborators is a technically well done study with all controls and expected experiments covered.  It is a significant contribution for anybody interested in the genetic dissection of a genomic region in any organism, as it covers the region of focus, genetic interval 57B on the 3rd chromosome in Drosophila comprehensively.  I fully and without hesitation recommend publication of this paper Genes. Please see my suggestion for a very minor revision below.

It would be helpful to the reader to have the presented mutagenesis affecting all genes in the 57B region studied summarized in a Figure.  I think figure 1 is  suited for that and can be easily expanded to cover that or simple summary figure added.  As it is shown now Figure 1 presents an overview of the region, showing the existing and new chromosome deficiencies and genes otp, DRx, hbn, cg34115 and Prosa3 with associated transposons and mtants. However, in addition to these genes the study describes the generation and partial analysis of CG15651, RIC-3, CG15649, CG15650, CG9344 and ND-B14.7 (all these were comprehensively mutated via a CRISPR targeting approach and Figs 4 and 5 are dedicated to that).  What should be added is how these additional genes are located/organized within the 57B region relative to the genes listed in Figure 1. 

Author Response

Answer to the reviewers

We are thankful to the anonymous reviewers for their time and effort spent with our manuscript. Their suggestions really improved the manuscript, and we hope that we have fulfilled all the suggestions with our modifications. We carefully revised the manuscript according to the reviewers’ comments. Alterations are indicated in red.

Answer to reviewer 1:

We thank you for your suggestion to expand figure 1, thereby providing the reader with a more comprehensive summary. We added section Fig.1B, showing an enlarged part of 57B covering the genes CG15651, RIC-3, CG15649, CG15650, CG9344 and NDB-B14.7, which were analyzed by gene targeting and CRISPR. The enlarged section was necessary, as with the original proportions of Fig.1A, the above-mentioned genes and their orientations wouldn´t have been visible. Therefore, the localization of the genes could be mentioned in the text, improving readers’ comprehension.

Kind regards, Eva Steinmetz and Uwe Walldorf

Reviewer 2 Report

Comments and Suggestions for Authors

In this study, the authors detailed the generation and identification of novel mutations within the specific 57B region of Drosophila melanogaster, employing both forward and reverse genetics methodologies. Through forward EMS mutagenesis screening, they successfully isolated and pinpointed an additional 36 mutations within the 57B region by examining 6278 offspring crosses. The team deciphered the genomic positions of three previously unidentified mutations via complementation testing, using known deficiency strains within this domain. Moreover, the researchers adeptly generated mutations in two established genes through the ends-out gene targeting technique and created mutations in four novel genes using the CRISPR/Cas9 system, all six of these genes reside within the 57B region. The culmination of this work has led to the isolation of gene mutations within this critical region, providing a valuable repository for future explorations into gene functionality.

To improve the paper more convincing, the following questions and concerns should be properly addressed:

  1. The assertion made by the authors that the genes CG9344 and CG15649 are non-lethal is based solely on visible strains bearing frameshift deletions or insertions, a conclusion that lacks convincing evidence. This claim is particularly unconvincing considering the inconsistency with results from RNAi experiments, where RNA interference targeting CG9344 and CG15649 induced early or late pupal lethality. Furthermore, the authors note potential off-target effects induced by the CRISPR/Cas9 system, raising the possibility that unintended edits in downstream genes or background mutations could negate the functionality of CG9344 and CG15649. To circumvent this issue, the outcross experiment should be performed to eliminate the background mutations, thereby ensuring a more reliable assessment of the phenotypes.

  1. The lethal phenotypes identified in the mutants of CG9344 and ND-B14.7 warrant validation through rescue experiments. This step is crucial to rule out potential influences from background mutations that may confound the interpretation of results. It is essential to confirm that the observed phenotypes are directly attributable to the mutations in question, not ancillary genetic alterations.

  1. Regarding the evaluation of mutation efficiency in the 57B region, the paper should specify the number of haplotypes analyzed during the EMS mutagenesis. Clarifying this detail would provide insights into the scope of the mutagenesis screen and the representativeness of the findings.

  1. In figure 5, the representation of CG15649 is ambiguous. It is critical to ensure the sequence is clearly depicted from the 5’ to 3’ end. This correction will aid readers in comprehending the information accurately.

  1. Figure 3, illustrating the generation of mutations by gene targeting, is overly simplistic and lacks informative detail. The authors should elaborate on the mechanism involved, specifically detailing how mutations were engendered. This includes a comprehensive description of the process including the white and loxp sites are flanked out through Cre recombinase activity. Providing a more detailed schematic representation or flow diagram could significantly enhance readers' understanding of the methodology and the genetic changes incurred.

Author Response

Answer to the reviewers

We are thankful to the anonymous reviewers for their time and effort spent with our manuscript. Their suggestions really improved the manuscript, and we hope that we have fulfilled all the suggestions with our modifications. We carefully revised the manuscript according to the reviewers’ comments. Alterations are indicated in red.

Answer to reviewer 2

ad 1.)

Thank you for reading thoroughly through our conclusions according to the RNAi and CRISPR experiments.

Data from CG9344 are consistent regarding the RNAi downregulations and the analysis of the CRISPR/Cas9 offspring. RNAi pointed to an early/late pupal lethality, and analysis of the CRISPR/Cas9 offspring showed alterations occurring outside or within the ORF, the latter also bearing a lethality that arises at the larval/pupal stage. So CG9344 was stated as a lethal gene according to the results of two independent methods.

According to CG15649 and CG15650 we show in our paper that some CRISPR mutants (that showed up homozygous viable during balancing process) have frameshift deletions in the respective genes around the transcription start without having a lethal effect. We therefore would argue that there are definitely no CRISPR off-target effects which lead to lethality in these two strains.

The from us also observed lethal strains (that showed up homozygous lethal during balancing process), showed no sequence alterations at the targeted CG15649 or CG15650 region. In these cases, we assume CRISPR off-targets that led to the lethality instead of a targeted indel mutation at the desired gene locus. Both observed results point to the complexity of the outcome of a CRISPR experiment and were therefore worth to be mentioned.

Concerning the RNAi strains targeting CG15649 and CG15650 which were made before our analysis by other people we are not so sure. Here it is not possible to rule out off-target effects in the RNAi experiments which might lead to lethal effects.

ad 2.)

Thank you for that advice. We fully agree with you, a rescue experiment would prove that the observed phenotypes are directly attributable to the mutations in CG9344 and ND-B14.7 and not from potential influences from background mutations. This kind of experiment would include the generation of corresponding transgenic flies first, as there is no appropriate strain available at the moment. As this would have exceed the aim and scope of our study we did not run through this experiment and furthermore, as the above-mentioned generation and following crossing experiments would take several months of time, we would not be able to manage the rescue experiment in a reasonable time frame.

To rule out the possibility that other mutations resulting from off-target effects are responsible for the lethality of the CG9433 and ND-B14.7 mutants, we did complementation test with our 57B deficiencies. In both cases we got no complementation with the deficiency used arguing that lethality is at least not resulting from off-target effects or background mutations outside the 57B region.

ad 3.)

We did our EMS mutagenesis screen 20 years ago according to the classical Drosophila screens by Eric Wieschaus and colleagues. The second chromosome was isogenized before the mutagenesis was done, but there were no genome sequence analyses or haplotype analyses performed.

ad 4.)

We would like to thank you for this advice. Indeed, it was depicted indefinitely. We improved the depiction of the sequences to leave no doubt in which orientation they are shown and where we defined CRISPR/Cas9 targeting regions. Now all wildtype (wt) sequences are shown double stranded and can be compared to the CRISPR-mutated strand(s) in below in an accurate way. Concerning CG15649 we presented sequence changes in the lower strand to make clearer what happened around the cleavage site and the PAM sequence. Which strand was chosen has been given by the target finder tool as described in our methods.

ad 5.)

Thank you for giving the constructive advice. To improve the understanding of the methodology and the genetic changes we followed your suggestions and made a flow diagram presenting the individual steps like the homologous recombination and the Cre recombination.

Kind regards, Eva Steinmetz and Uwe Walldorf

Round 2

Reviewer 2 Report

Comments and Suggestions for Authors

The authors have addressed all my comments, and the manuscript is well improved and ready for publication.